# Bacterial Adhesion on Prosthetic and Orthotic Material Surfaces

**Anže Abram** [1] **, Anamarija Zore** [2] **, Urban Lipovž** [2] **, Anita Košak** [2] **, Maja Gavras** [1] **, Žan Boltežar** [2]
**and Klemen Bohinc** [2],*

1   Jozef Stefan Institute, 1000 Ljubljana, Slovenia; anze.abram@ijs.si (A.A.); maja.gavras@ijs.si (M.G.)
2   Faculty of Health Sciences, University of Ljubljana, 1000 Ljubljana, Slovenia; anamarija.zore@zf.uni-lj.si (A.Z.);
    ulipovz@gmail.com (U.L.); anita.kosak@gmail.com (A.K.); zan.boltezar@zf.uni-lj.si (Ž.B.)
*   Correspondence: klemen.bohinc@zf.uni-lj.si

**Abstract:** Prosthetic and orthotic parts, such as prosthetic socket and inner sides of orthoses, are often in contact with human skin, giving bacteria the capability to adhere and form biofilms on the materials of those parts which can further cause infections. The purpose of this study was to determine the extent of bacterial adhesion of *Staphylococcus aureus* and *Staphylococcus epidermidis* on twelve different prosthetic and orthotic material surfaces and how roughness, hydrophobicity, and surface charge of this materials affect the adhesion. The roughness, contact angle, zeta potential of material surfaces, and adhesion rate of *Staphylococcus aureus* and *Staphylococcus epidermidis* were measured on all twelve prosthetic and orthotic materials, i.e., poly(methyl methacrylate), thermoplastic elastomer, three types of ethylene polyvinyl acetates (pure, with low-density polyethylene and with silver nanoparticles), silicone, closed-cell polyethylene foams with and without nanoparticles, thermo and natural cork, and artificial and natural leather. The greatest degree of adhesion was measured on both closed-cell polyethylene foams, followed by artificial thermo cork and leather. The lowest adhesion extent was observed on ethylene-vinyl acetate. The bacterial adhesion extent increases with the increasing surface roughness. Smaller deviations of this rule are the result of the surface's hydrophobicity and charge.

**Keywords:** bacterial adhesion; surface characterization; prosthetic and orthotic material surfaces





## 1. Introduction

Prostheses and orthoses are mobility assistive devices that are often in touch with the user's skin. Specifically, prosthetic and orthotic parts that are usually in direct contact with the user's skin are inner sides of prosthetic sockets and inner sides of orthotic devices [1]. The skin acts as a physical barrier that protects our body against possible foreign microorganisms. It is inhabited by microbiota-always present microorganisms, such as bacteria, fungi, and viruses [2]. The most well-known bacteria that inhabit the skin's permanent microbiota are staphylococci, most notably *Staphylococcus (S.) aureus* and *S. epidermidis* [3]. Although they are a commensal on the human skin, some strains can effectively adhere to solid surfaces and develop biofilms that can be a leading cause of infections when coming in contact with the impaired skin barrier [4].

Bacterial adhesion to the surface occurs when the bacteria are firmly attached to the surface [5]. Process of bacterial adhesion to the surface of materials is often described as nonspecific and specific. The first phase involves nonspecific adhesion, covered by the physicochemical interactions between the material's surface and bacterial cell wall. These interactions include Van der Waals forces, electrostatic forces, and hydrophobic interactions and represent the initial degree of adhesion, where adhesion is still reversible [6]. In the second phase of the adhesion process, interactions between material surfaces and surface structures of bacteria occur. These bacterial surface polymer structures are comprised of fimbria, capsules, and mucus. They consist of polysaccharides and proteins that act as bacterial adhesins [7].

Factors found to affect bacterial adhesion include surface hydrophobicity, cell surface charge, and roughness of the material surface [8]. Bacteria with hydrophobic properties are attached more strongly to surfaces of hydrophobic materials. In addition, hydrophobic bacteria's adhesion is greater than the adhesion of hydrophilic bacteria. In general, hydrophobic materials showed less resistance to bacterial adhesion than hydrophilic materials [9]. Surface roughness is also a relevant factor to be considered in the process of adhesion [10]. It represents a part of the defects on the surface (dimples, grooves, and cracks) that may result from the manufacturing process or the wear [11]. Greater roughness of the materials' surface generally encourages bacterial adhesion and, therefore, the development of biofilm [12,13]. Most solid surfaces and bacteria are negatively charged [14]. If bacteria and materials surface have the same charge, electrostatic forces will be repulsive between the bacteria and the material's surface. This repulsion causes an energy barrier to adhesion [15]. In addition, some specific surface coatings can influence the bacterial adhesion [16].

Orthotic and prosthetic parts are made of various materials. Traditional prosthetic sockets are made of hard materials [17], commonly made of laminated plastic material in full contact over the entire surface of the residual limb. The material chosen for the study is poly(methyl methacrylate) (PMMA) or acrylic resin, one of the most commonly used materials for such applications [18]. Other sockets can be flexible, supported by a rigid frame, which ensures comfort while walking and sitting [17]. Such socket is made of an elastic thermoplastic material in full contact with the residual limb and allows for the muscle contraction and expansion during walking [19]. Example of these materials in this study are ethylene-vinyl acetate (EVA) and a material that is a mixture of EVA and low-density polyethylene (LDPE). In some cases, the socket surface is touching the skin, whilst, in other cases, the inserts or liners may be used between the residual limb and the hard socket [20]. These interface materials control the forces and increase the prosthesis's comfort. They can also provide a better suspension to the residual limb [21]. A silicone liner, a gel liner from thermoplastic elastomer (TPE), and closed-cell polyethylene were used in the study. Cork and leather are frequently used materials in orthotic application, especially in orthotic footwear [1,22]. Natural and artificial leather, as well as natural cork and thermo-cork, were used in this study. Because these materials can be in direct contact with the skin, a bacterial cell on the skin comes close to the material's surface, where bacterial adhesion can occur [23].

Most of the studies about the bacterial adhesion to medical devices are primarily concerned with the adhesion of staphylococci to materials associated with implantable medical devices [8,24], but no studies describing the adhesion of bacteria on prosthetic and orthotic materials were found. The paper aims to demonstrate how the surface properties of various prosthetic and orthotic materials that are in contact with the skin, such as surface roughness, hydrophobicity, and surface charge, impact the extent of *S. aureus* and *S. epidermidis* adhesion onto the surface.

## 2. Materials and Methods

### 2.1. Materials for the Prosthesis and Orthosis

Twelve different materials commonly used in prosthetics and orthotics were selected in this study (Table 1). Out of the twelve, two materials were natural materials: natural leather (Walkleder-Hälse, Ortho-Reha Neuhof, Nürnberg, Germany) and natural cork (Flexo-Kork, Ortho-Reha Neuhof, Nürnberg, Germany), while the other ten were synthetic materials. Three materials had coating of silver (Ag) nanoparticles that are used for their antibacterial properties: EVA with Ag NP (Anitbacterial ThermoLyn, Ottobock, Duderstadt, Germany), closed-cell polyethylene with Ag NP (Pedilin SilverShield, Ottobock, Duderstadt, Germany), and EVA/LDPE (ThermoLyn EVA/LDPE SilverShield, Ottobock, Duderstadt, Germany). ThermoLyn EVA/LDPE SilverShield is composed of 24% EVA and 75% LDPE.

**Table 1.** List of materials tested.

| Material | Product Name | Manufacturer |
|---|---|---|
| Ethylene-vinyl acetate (EVA) | ThermoLyn soft | Ottobock |
| Ethylene-vinyl acetate (EVA) with Ag NP | Anitbacterial ThermoLyn | Ottobock |
| Closed-cell polyethylene (PE) | Pedilin | Ottobock |
| Closed-cell polyethylene (PE) with Ag NP | Pedilin SilverShield | Ottobock |
| Natural cork | Flexo-Kork | Ortho-Reha Neuhof |
| Thermo-cork | Thermo-Kork | Ortho-Reha Neuhof |
| Natural leather | Walkleder-Hälse | Ortho-Reha Neuhof |
| Artificial leather | Kunstleder mit Gewebe | Ortho-Reha Neuhof |
| Silicone | Iceross Seal-In® X TF | Ossur |
| Thermoplastic elastomer (TPE) gel | Alpha Hybrid® Liner | WillowWood |
| Ethylene-vinyl acetate/low-density polyethylene (EVA/LDPE) with Ag NP | ThermoLyn EVA/LDPE SilverShield | Ottobock |
| Poly(methyl methacrylate) (PMMA) | Lamineirheartz C | Ortho-Rhea Neuhof |

To measure the bacterial adhesion, surface roughness, and water contact angle, each material was cut to the dimensions of 2 cm × 2 cm, and 1 cm × 2 cm to measure the zeta potential. Due to the greater elasticity, silicone and TPE were glued to a piece of 1-mm polyethylene plate and cut to the desired dimension.

### 2.2. Bacteria

*S. aureus* and *S. epidermidis* are often the part of the skin microbiota. They are facultative anaerobes and, especially *S. aureus*, can infect almost any tissue in the body, frequently the skin. *S. epidermidis* is a frequent contaminant of catheters and surgical implants, where it forms biofilms. Both bacteria are often involved in nosocomial infections. In the study, the standard isolate *S. aureus* ATCC 25923 and *S. epidermidis* isolated from clinical specimen was used and obtained at the Institute for Microbiology and Immunology, Medical Faculty, University of Ljubljana. Both bacteria are Gram-positive cocci (round-shaped), and form grape-like structures.

Bacterial strains were applied on blood-agar plates and were incubated for 24 h at aerobic conditions at 37 °C. From blood-agar cultures, overnight culture was prepared and diluted in 1:30 ratio. Samples of prosthetic and orthotic materials were then incubated in this suspension. Samples were before incubation sterilized with UV light and then used for cultivation with diluted overnight culture for 14 h at 37 °C regarding to the growth curve of both bacteria. After incubation, samples were rinsed three times with PBS and three times with water.

### 2.3. Roughness

Surface topography was assessed by mechanical profilometer Form Talysurf Series 2 (Taylor-Hobson Ltd., Leicester, UK). The mechanical profilometer uses a probe which is physically moving along the surface in order to acquire the surface height. Three 5-mm-long parallel measurements were taken across the samples' surface. DIN EN ISO 4288:1998 and Gaussian cut-off filter 0.25 mm were used to obtain topographic values, such as the arithmetic average roughness $R_a$ and root mean square (RMS) roughness $R_q$.

### 2.4. Contact Angle

Wetting of the surface was assessed by the sessile drop method on the Theta Optical Tensiometer (Attension, Stockholm, Sweden), which consisted of a light source, camera, liquid dispenser, and a sample stage. We put a water droplet on the prosthetic material

surface and measured the contact angle between the droplet and the prosthetic surface. A series of 5 measurements were performed on each substrate. A quick sequence of shots was taken on porous samples which quickly absorb the droplet.

### 2.5. Streaming Potential

Streaming potential measurements were performed on Anton Paar SurPASS™ 2 Electrokinetic Analyzer. First, 1 mM Phosphate-buffered saline (PBS) was used as an electrolyte at pH 7.4 and room temperature for four repetitions per sample in the adjustable-gap cell. The zeta potential $\zeta$ was then calculated from the measured streaming potential [25].

### 2.6. Monitoring the Adhesion Extent

Scanning electron microscopy (SEM) was used to assess the surfaces' bacterial coverage and morphology. A 7-nm-thick Au layer was applied with GATAN Model 682 PECS system (Precision Ion Etching and Coating System, GATAN Inc., Pleasanton, CA, USA) to obtain a conductive surface. The samples' qualitative and quantitative analysis was performed on JEOL JSM-7600F (JEOL, Tokyo, Japan). The preferential location for bacterial adhesion was visually assessed qualitatively. Quantitative analysis was performed on contrasted SEM micrographs by encircling and converting the images to binary form. The coverage of *S. aureus* and *S. epidermidis* was calculated using ImageJ software package (Version 1.50b, 2015, Wayne Rasband, National Institutes of Health, Bethesda, MD, USA). Several images, representing a total area of approximately 500 $\mu m^2$, were analyzed. Conventional bacteria counting techniques, such as dye staining, have proven to be unsuitable due to sample porosity, rough surfaces, and leeching of dye into the substrate.

## 3. Results

### 3.1. Roughness

RMS roughness $R_q$ for all materials considered is shown in Figure 1. The highest roughness was measured on Closed-cell PE foam without Ag nanoparticles (17.03 $\pm$ 1.87) $\mu m$ and with Ag nanoparticles (16.17 $\pm$ 1.32) $\mu m$. Other surfaces with the roughness in the micrometer range are thermo (10.06 $\pm$ 0.22) $\mu m$ and natural (3.12 $\pm$ 0.43) $\mu m$ corks, as well as natural (5.83 $\pm$ 0.60) $\mu m$ and artificial (8.77 $\pm$ 1.11) $\mu m$ leathers. The lowest roughness below micrometer was measured on EVA surfaces (~0.02 $\mu m$ with and without Ag nano particles), EVA/LDPE with Ag nano particles (0.48 $\pm$ 0.08) $\mu m$, silicone (0.45 $\pm$ 0.12) $\mu m$, and PMMA (0.17 $\pm$ 0.09) $\mu m$.

### 3.2. Contact Angle

Figure 2 represents the contact angle measurements. The most hydrophobic behavior was observed on closed-cell PE foam with and without Ag nanoparticles (contact angles of 130.36° $\pm$ 3.20° and 122.23° $\pm$ 1.38°, respectively), followed by Thermo and natural cork (contact angles of 112.94° $\pm$ 4.43° and 109.22° $\pm$ 4.17°, respectively) and silicone (103.05° $\pm$ 5.13°). The hydrophilic behavior was observed on EVA with and without NPs (86.10° $\pm$ 0.64° and 89.23° $\pm$ 0.89°, respectively), EVA/LDPE + Ag NP (59.50° $\pm$ 1.78°), PMMA (80.11° $\pm$ 6.32°), TPE (59.44° $\pm$ 4.10°), and artificial leather (69.49° $\pm$ 2.68°). Natural leather absorbed water droplets due to its porous structure, and, consequently, the surface's wetting characteristics instantaneously changed.

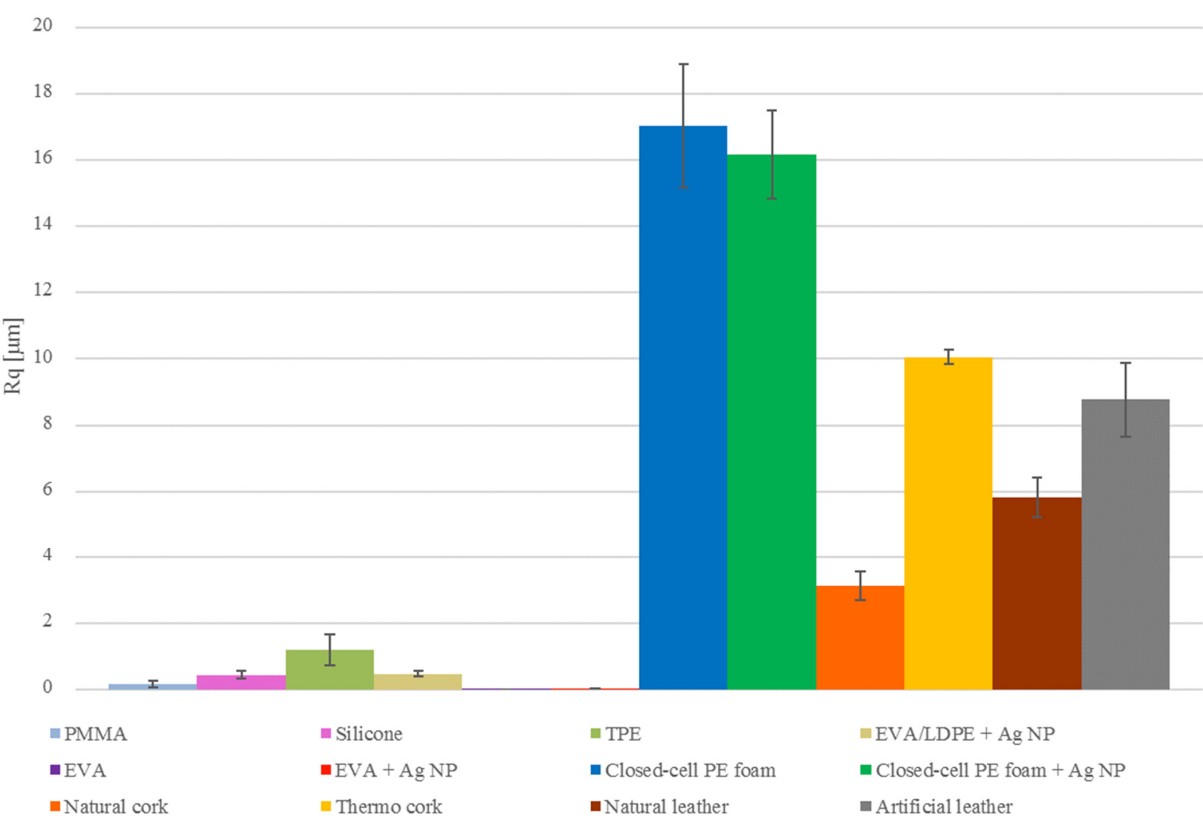

**Figure 1.** RMS roughness R$_q$ of orthotic and prosthetic surfaces measured with the profilometer.

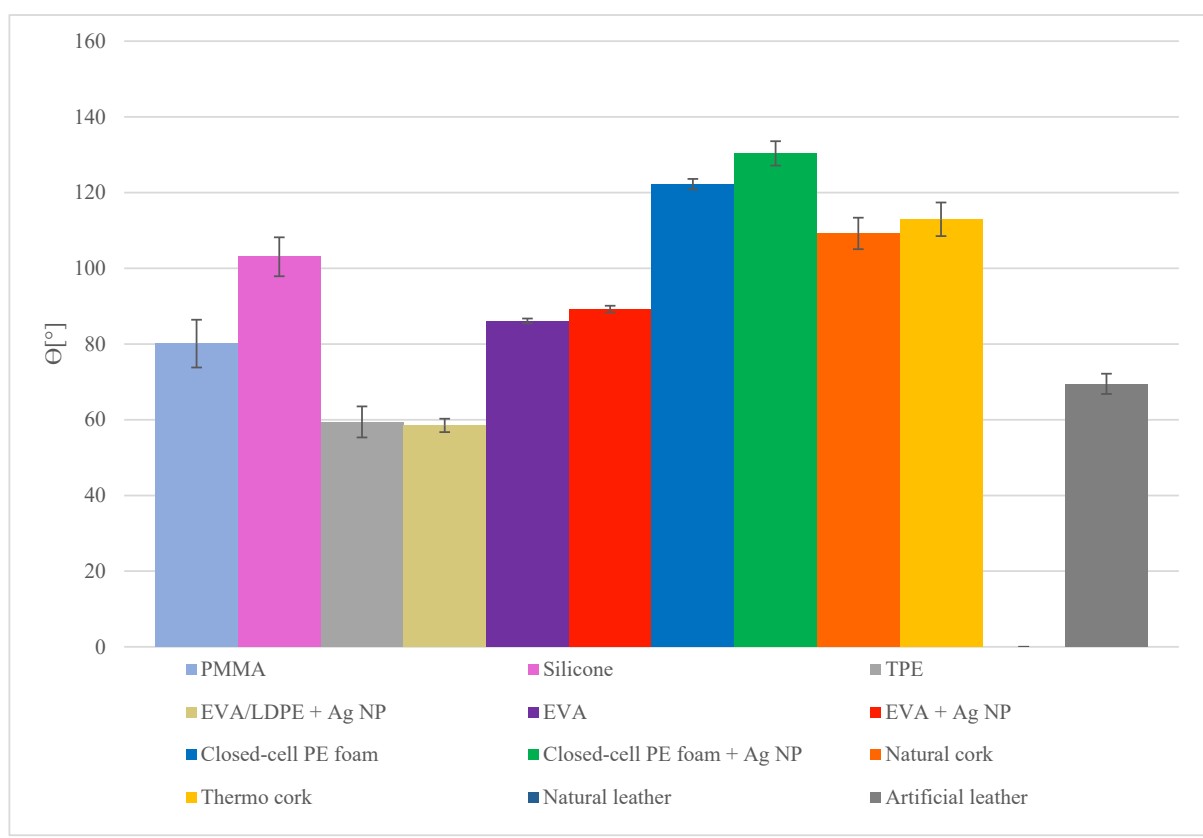

**Figure 2.** Water contact angles on different orthotic and prosthetic material samples measured with the tensiometer.

### 3.3. Zeta Potential

The zeta potential in phosphate-buffered saline solution at pH 7.4 of all tested materials was negative (Figure 3). Artificial leather had the most negative potential ($-121.57 \pm 3.53$) mV, followed by silicone ($-91.77 \pm 0.90$) mV and PMMA ($-61.64 \pm 9.08$) mV and EVA with and without Ag nanoparticles ($-48.71 \pm 1.34$) mV and ($-44.63 \pm 2.31$) mV, respectively). EVA/LDPE with Ag NP has the zeta potential ($-33.85 \pm 1.31$) mV. Similarly, closed-cell PE foam surfaces with Ag nanoparticles had more negative zeta potential than the surface without Ag nanoparticles ($-27.19 \pm 0.24$) mV and ($-20.58 \pm 0.70$) mV, respectively). The least negative zeta potential was measured on the thermo cork surface ($-17.84 \pm 0.32$) mV. The streaming potential measurements of natural cork and leather were unreliable due to the porosity of the surfaces with very good wetting properties and unappropriated measuring cell.

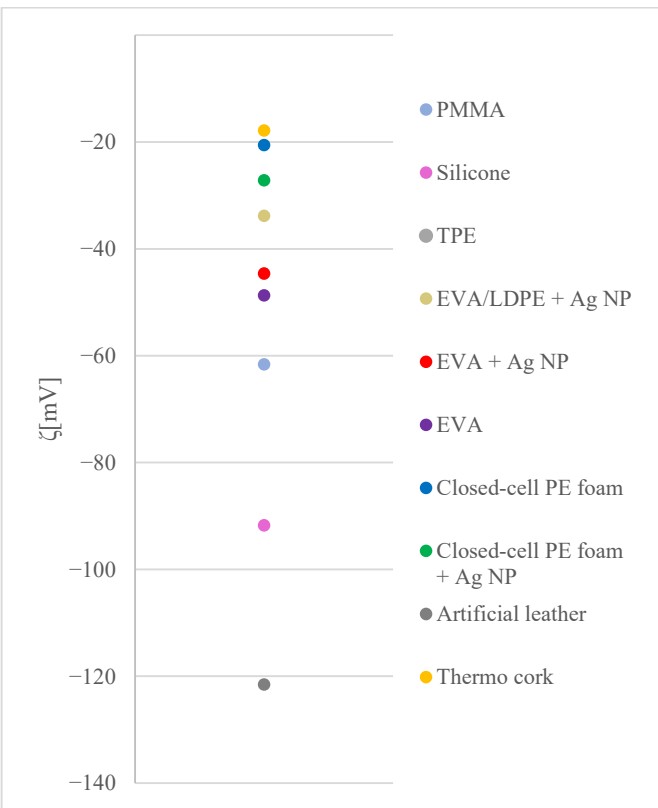

**Figure 3.** Zeta potentials of orthotic and prosthetic material surfaces.

### 3.4. Growth Curves

Figure 4a shows the growth curve of *S. epidermidis*, i.e., the time dependence of the colony-forming units per milliliter. Bacterial growth reached the peak after 15 h of incubation at 37 °C. The peak in the curve corresponded to approximately $1.6 \times 10^9$ CFU/mL of culture. Figure 4b shows the growth curve data of *S. aureus* as measured in Bohinc et al. [26]. The bacterial growth reached the peak after 13 h of incubation at 37 °C. The peak in the curve corresponded to approximate absorbance of 0.13.

### 3.5. Bacterial Adhesion Rate

Figure 5 shows the bacterial adhesion extent on orthotic and prosthetic materials. The extent was determined by the percentage of the surface covered by bacteria, which was assessed from SEM micrographs (Figures 6–8). The assessment of bacterial coverage from SEM micrographs is statistically inferior due to relatively smaller sampling area which reflects greater statistical error. In addition, a quite strong coverage dispersion between

different areas on the same sample was observed. The greatest bacterial adhesion extent was obtained on Closed-cell PE foam with Ag NP (61.6% surface coverage), followed by artificial leather with 40.3%. Regarding bacterial extent, very close to artificial leather are thermal cork and natural leather. All EVA surfaces, PMMA, silicone, TPE, and natural cork have one order of magnitude lower bacterial adhesion extent than the materials with the maximal extent.

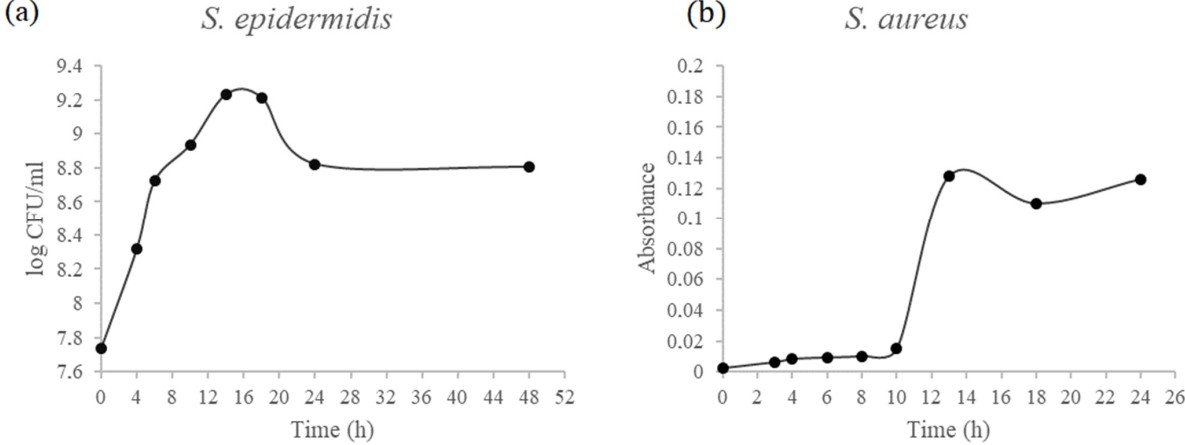

**Figure 4.** Growth curves of *S. epidermidis* (**a**) and *S. aureus* (**b**) in nutrient broth. For *S. epidermidis*, colony-forming unit per milliliter (CFU/mL) as a function of time is shown, whereas, for *S. aureus*, the time dependence of absorbance is presented [26]. Reprinted with permission from [26]. Copyright © 2014 Elsevier.

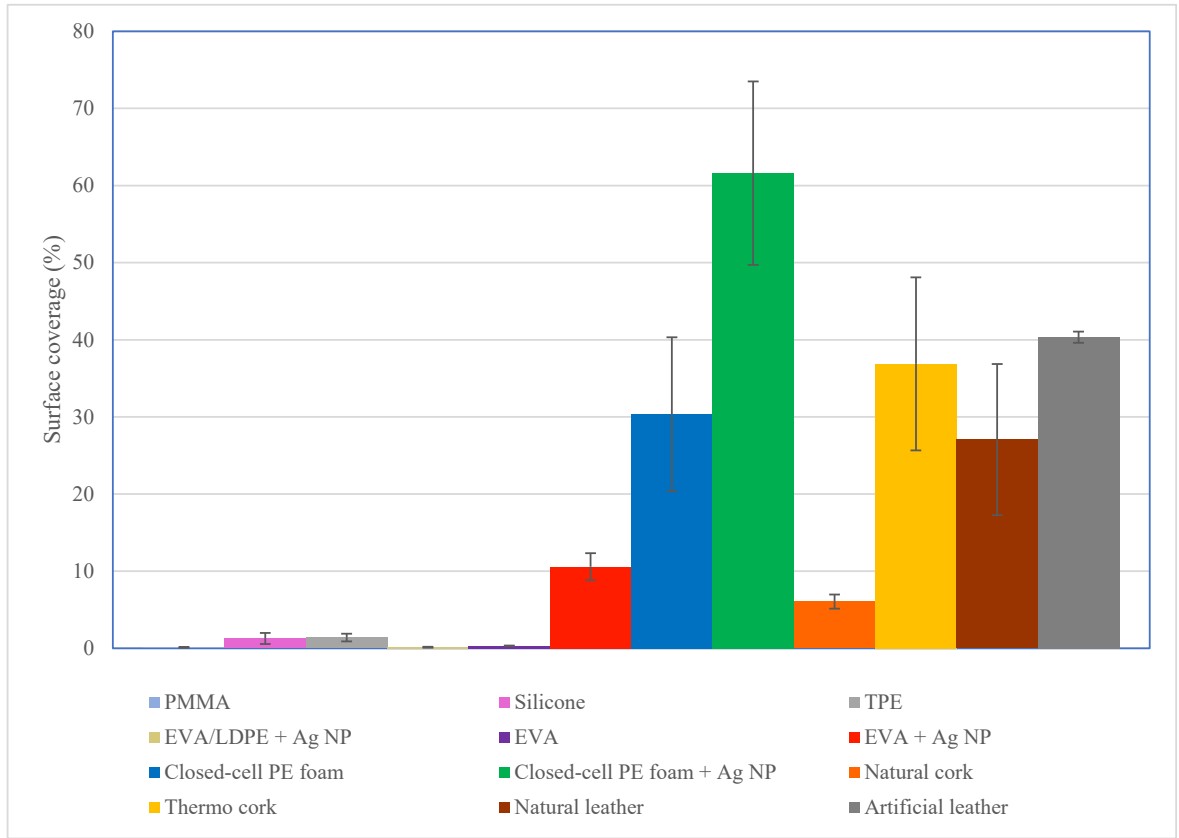

**Figure 5.** Surface coverage for different orthotic and prosthetic materials based on the evaluation of SEM micrographs. On the first four surfaces, *S. epidermidis* is adhered, whereas, on the rest of the surfaces, *S. aureus* is adhered.

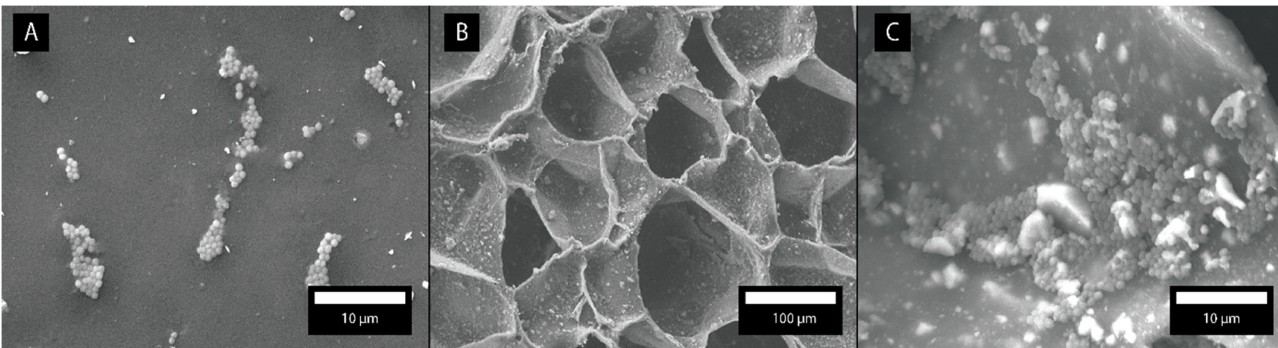

**Figure 6.** SEM micrographs of (**A**) EVA flat surface with bacteria forming small flakes around asperities. PE and cork samples have similar honeycomb structure (**B**). Bacteria are mostly adhered to the well walls in the form of closely packed films (**C**).

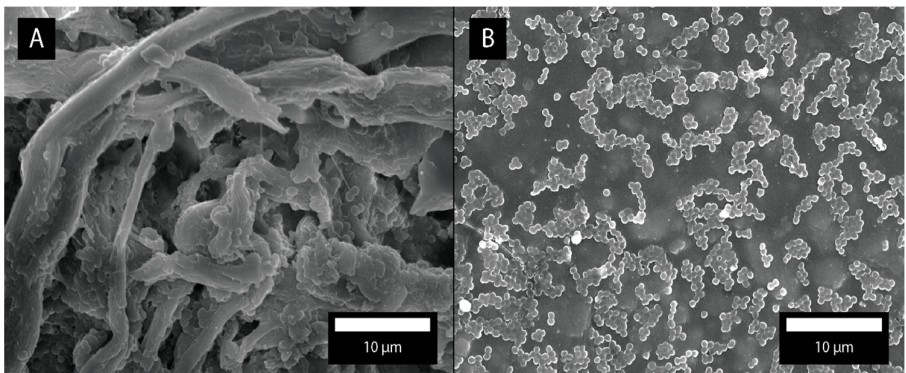

**Figure 7.** SEM micrographs of natural (**A**) and artificial leather (**B**). The difference in the surface morphology dictates the bacterial adhesion.

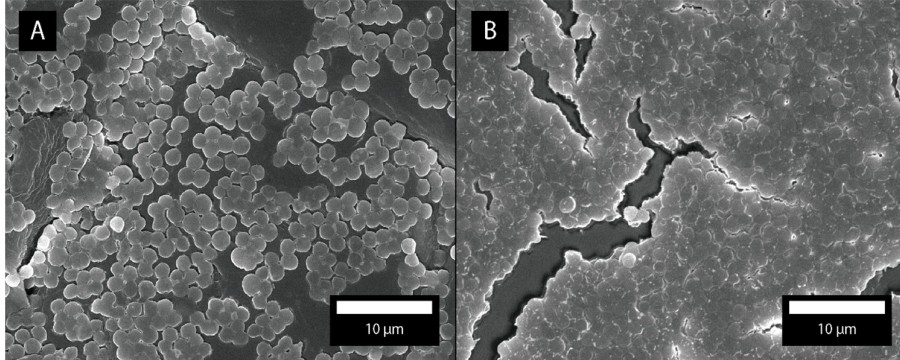

**Figure 8.** Closed-cell PE surface (**A**) with Ag nanoparticles and (**B**) without Ag nanoparticles. Most of the surface is covered with spherical *S. aureus*.

Micrographs in Figure 6 clearly show the diversity of the sampled materials' surface morphologies. Bacteria preferentially form "flakes" on flat surfaces. Bacteria stick to grooves, pits, and generally more curved asperities, where they form non-linear colonies.

SEM micrograph of natural leather (Figure 7A) is distinctive among the samples due to its fiber structure. Bacteria are predominantly adhered to pits and valleys in between the fibers. Artificial leather surface exhibits no distinctive features for preferential bacterial adhesion. Therefore, adhered bacteria are evenly distributed over the artificial leather surface (Figure 7B).

Micrographs in Figure 8 show the surface of closed-cell PE samples with (A) and without (B) Ag nanoparticles. The bacterial coverage of the closed-cell PE surface with Ag nanoparticles is 80.3%, whereas, on the surface without Ag nanoparticles, the bacterial coverage is 90.2%.

Micrographs in Figure 9 show the prosthetic material surfaces: PMMA, EVA/LDPE with Ag nanoparticles, silicone, and TPE. The bacterial coverage of *S. epidermidis* to prosthetic materials is very low compared to most prosthetic materials covered with *S. aureus*. Among prosthetic materials, the largest bacterial adhesion was obtained for silicone and TPE.

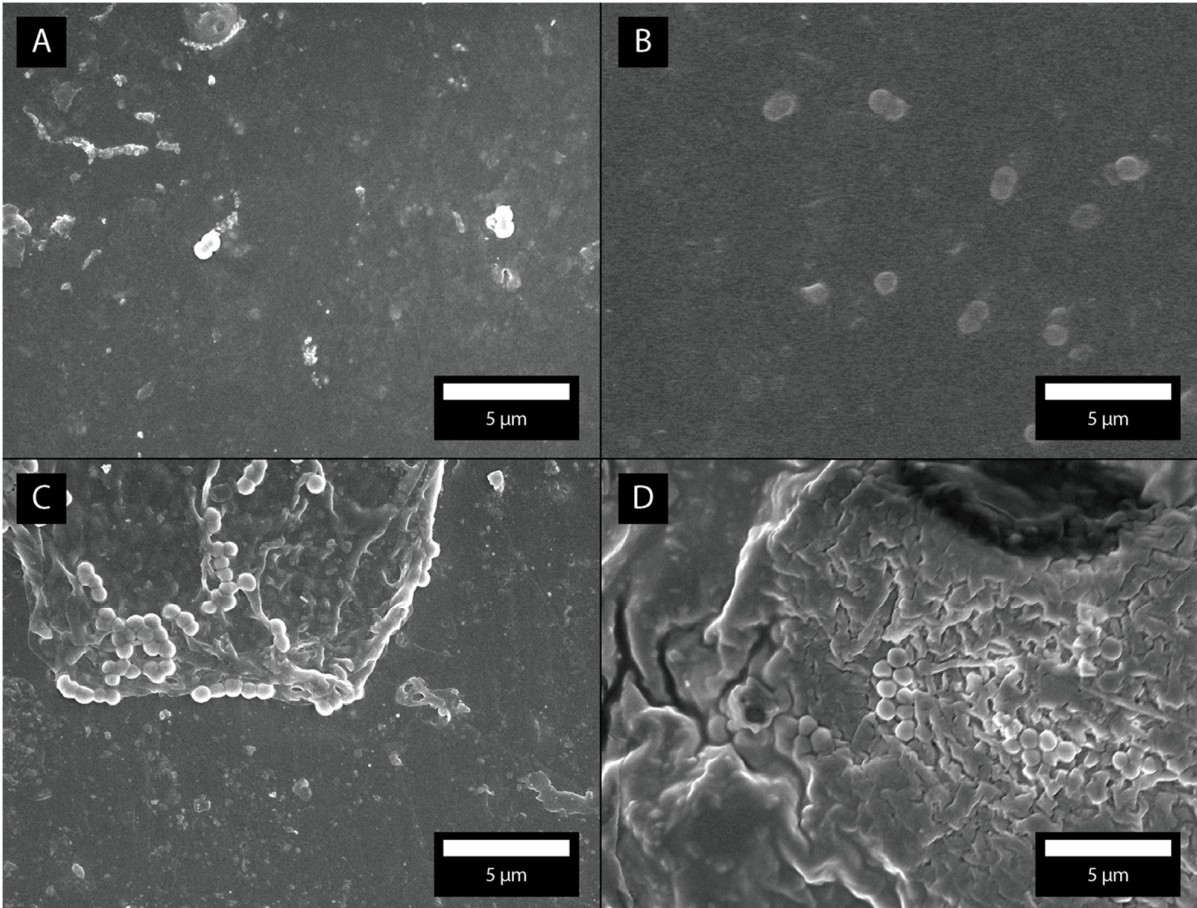

**Figure 9.** SEM micrographs of (**A**) PMMA, (**B**) EVA/LDPE with Ag nanoparticles, (**C**) silicone, and (**D**) TPE surfaces with adhered *S. epidermidis*.

## 4. Discussion

Parts of prostheses and orthoses usually consist of various materials which are in contact with the skin. On a daily basis, orthotics and prosthetics practitioners are faced with the decision to choose a material most suitable for a specific application. Besides good mechanical properties, the orthotic and prosthetic materials must also have good antibacterial properties. Such materials minimize the chance for the infection [1]. In the present work, we evaluated the most used orthotic and prosthetics materials, for their ability to attract bacteria commonly present on the skin, such as *S. aureus* and *S. epidermidis* [27]. The study focused on the surface characteristics (wetting, surface charge, roughness, and morphology) of the materials used since these, along with the characteristics of specific bacteria itself, predominantly influence the interaction of bacteria with the material surface [26]. Moreover, we evaluated how coatings of Ag nanoparticles affect the bacterial adhesion to materials.

Research, related to the bacterial adhesion to natural and artificial orthotic and prosthetic materials, is scarce. Studies of *S. aureus* and *S. epidermidis* adhesion to material surfaces are almost exclusively related to titanium and titanium alloys, used as implants, where the dynamics of biofilm formation are of utmost importance [28]. However, the knowledge collected has a limited application to the field of orthotics and prosthetics.

Bacterial adhesion to the surface occurs when the bacteria are firmly attached to the surface [5]. The first phase involves non-specific adhesion where Van der Waals forces, electrostatic forces, and hydrophobic interactions between material surface and bacterial cell wall represent the initial degree of adhesion, where adhesion is still reversible. At long distances (>10 nm), the attractive forces (Van der Walls forces) are larger than the repulsive forces (electrostatic forces), resulting in the attraction between charged surfaces and particles [29]. At this distance, attractive forces are in the so-called secondary minimum of potential energy and are weak. At shorter distances, the repulsive forces are increasing according to attractive forces. There is strong repulsion in the maximum energy between the two bodies. If the repulsive forces can be defeated at very short distances (1 nm), there is mutual attraction (the primary minimum of potential energy). The attraction between the two particles is strong at this distance and becomes irreversible [6]. At the short-range distances, hydrogen bonds, ionic dipole interactions, and hydrophobic interactions occur [30]. The second adhesion phase consists of specific interactions between material surfaces and bacterial surface structures, such as capsules, fimbria, and mucus, which consist of polysaccharides and proteins that act as bacterial adhesins [5]. Fimbria (pili) are structures on the bacterium's surface, which consist of protein subunits [31]. Capsules and mucus are part of extracellular polymeric substances. According to their structure, they are polysaccharides, also known as glycocalyx (hydrated polymeric mucous matrix), which, in turn, allow the bacteria to encapsulate on the surface and cause the formation of biofilm [5,32].

Surface asperities positively affect bacterial adhesion, providing the shelter for lateral forces and strong attachment point [27]. EVA samples show the lowest roughness of tested materials and, consequently, exhibited the lowest bacterial adhesion, by orders of magnitude compared to other materials. As shown in Figure 6, the surfaces significantly differed in morphology which cannot be fully characterized by a single roughness value. The topography of honeycomb structure in PE foams and cork (Figure 6) and fiber-like morphology of natural leather (Figure 8) cannot be fully determined with stylus-type profilometry measurements. More precise methods, such as the atomic force microscopy (AFM), could not be used because the morphology is not flat enough to perform the measurements. Gaps, pits, and valleys on those materials provide shelter and attachment points for bacteria.

Figure 3 shows the zeta potential of materials used. Using the Grahame equation, the surface charge density was obtained from the measured zeta potential. The surface charge significantly affects bacterial adhesion to the surface. There is a strong electrostatic attraction between the oppositely charged bacterium and surface. Both bacterial strains, *S. aureus* and *S. epidermidis*, are negatively charged and positively charged or weakly negatively charged surfaces are more prone to the bacterial adhesion [27,33,34]. Prosthetic and orthotic surfaces in this study are negatively charged. The most negative surface charge was measured on artificial leather (−121.57 ± 3.53) mV, which was also one of the surfaces with the most adhered bacteria. Streaming potential measurements proved to be problematic for foamy and otherwise porous materials, so only a handful of materials were characterized this way. Generally, no apparent connection between surface charge and bacterial adhesion has been observed due to our samples' negative nature.

Besides surface roughness [35] and zeta potential [34], surface hydrophobicity can also be a driving force for bacterial adhesion. The highest hydrophobicity was measured on Closed-cell PE foam, whereas the lowest hydrophobicity was obtained on the silicone surface. Similar observations on silicone were found by Lin et al. [36] and on the polyzwitterion/Enzyme coatings on silicone [37].

Due to the uneven and highly diverse surfaces (porosity, surface features), the SEM micrographs were chosen over dye staining methods to ensure that the correct number of bacteria was counted on each micrograph and preferential attachment points evaluated at the same time. The use of staining dyes and optical bacterial counting was further complicated due to the heavy background staining. In Figure 5, surface coverage is shown for different materials. The comparison of surface roughness course with bacterial adhesion extent reveals that the leading factor which affects the bacterial adhesion is the roughness. This means that with increasing roughness the bacterial adhesion extent also increases.

The study also evaluated Ag nanoparticles' effect on the antibacterial properties of materials. No definite conclusion can be drawn here. Traditional methods for bacteria counting by live/dead staining dyes and optical microscopy yielded unsatisfactory results due to the dye leaching into the substrate and general roughness and morphology of chosen materials. Therefore, the counting from SEM micrographs has been employed, which is more precise on a smaller scale, but, on larger scale, it is statistically inferior to optical methods. Ag nanoparticles are positively charged, which might not be shown on streaming potential measurements of the bulk but can positively influence the local bacterial attachment from which the biofilm grows. However, micrographs in Figure 8 show the different nature of bacterial grown on the samples with and without Ag nanoparticles. The bacteria on the sample without the Ag nanoparticles are much more closely packed and appear to grow in layers, while the bacteria on Ag-coated sample grow in a single layer and are not tightly packed. The bacteria's integrity (live or dead) from SEM micrographs could not be assessed in this study.

## 5. Conclusions

This study investigated the impact of different prosthetic and orthotic materials on bacterial adhesion of *S. epidermidis* and *S. aureus*. To better understand the bacterial adhesion, the surface topography, roughness, hydrophobicity, and zeta potential of all materials under consideration were measured. From the SEM micrographs, the bacterial adhesion extent was determined. This study showed that the highest bacterial adhesion was on the artificial leather and thermo cork, whereas lowest adhesion extent was observed on ethylene-vinyl acetate. The reason lies in the specific material properties and surface's characteristics, such as roughness, hydrophobicity, and charge.

This study helps in understanding which orthotic and prosthetic material reduce bacterial adhesion when exposed to the bacterial flora. The results can help professionals in prosthetics and orthotics to select the most appropriate materials regarding bacterial adhesion and can, therefore, reduce the risk of bacterial infection in the prosthetic itself and orthotic users.

**Author Contributions:** Conceptualization: K.B., Data curation A.A., A.Z., M.G., U.L. and A.K. Investigation K.B., A.A., A.Z., M.G., U.L. and A.K. Writing original draft: A.A. and K.B. Writing review and editing Ž.B. All authors have read and agreed to the published version of the manuscript.

**Funding:** This research received no external funding.

**Institutional Review Board Statement:** Not applicable.

**Informed Consent Statement:** Not applicable.

**Data Availability Statement:** Data available by corresponding author on request.

**Acknowledgments:** Authors thank Moor d.o.o. for providing orthotic and prosthetic materials.

**Conflicts of Interest:** The authors declare no conflict of interest.

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
