# Peer review of "Bacterial Adhesion on Prosthetic and Orthotic Material Surfaces"

_coatings, doi:10.3390/coatings11121469_

Round 1

Reviewer 1 Report

In this contribution by Abram and co-workers, the authors discuss bacterial adhesion on prosthetic and orthotic material surfaces. The results are attractive to the readership of coatings. However, the following points should be solved before it could be considered for the publication.

  1. Line 45, remove ‘hydrophobicity surface charge’.
  2. Line 263-266, at long distances (larger than 10 nm), usually Van der Walls (VdW) forces are weaker than the electrostatic forces but not larger. When the distance between two similar charged surfaces becomes smaller than 10 nm, VdW starts to be larger than electrostatic forces, and a ‘jump-in’ between the charged surfaces may happen, which is described in details in a recent review (doi.org/10.1016/j.cocis.2019.10.001). The ref. is encouraged to be included.
  3. In the discussion, several studies (doi.org/10.1016/j.actbio.2011.02.001; 10.1021/acsnano.9b09282; 10.1021/acs.langmuir.5b04303) related to antibacterial adhesion surfaces should be included.
  4. Formats issue. Line 12-13, ‘hydrophobicity surface charge’ to ‘hydrophobicity, surface charge’. Check all.

Author Response

Reviewer 1:

  1. Line 45, remove ‘hydrophobicity surface charge’.

Corrected the typo.

  1. Line 263-266, at long distances (larger than 10 nm), usually Van der Walls (VdW) forces are weaker than the electrostatic forces but not larger. When the distance between two similar charged surfaces becomes smaller than 10 nm, VdW starts to be larger than electrostatic forces, and a ‘jump-in’ between the charged surfaces may happen, which is described in details in a recent review (org/10.1016/j.cocis.2019.10.001). The ref. is encouraged to be included.

The reference was included.

  1. In the discussion, several studies (doi.org/10.1016/j.actbio.2011.02.001; 10.1021/acsnano.9b09282; 10.1021/acs.langmuir.5b04303) related to antibacterial adhesion surfaces should be included.

Above-mentioned studies have been considered and included in the manuscript.

  1. Formats issue. Line 12-13, ‘hydrophobicity surface charge’ to ‘hydrophobicity, surface charge’. Check all.

Formatting issue corrected.

Reviewer 2 Report

Dear authors,

Xe have read with interest your manuscript investigating bacterial adhesion on various materials for intra oral applications.

The topic/problematic is well introduced, and methods are clearly described and consistent with the objectives of the work.

Results are clear, and discussion/conclusion is supported by the experimental data.

The main weakness is, finally, that the obtained results are (too) logical: roughness increases bacterial adhesion.

We have one remark concerning results presentation: histograms and graphs are not nicely presented and could be improved. Y axis legend and value (12,000µm should be 12µm or 8,4 should be 8.4), or unnecessary horizontal lines, or absence of significant difference indications over histograms...

Therefore we recommend your manuscript to be accepted for publication after these modifications.

Yours faithfully

Author Response

Reviewer 2:

  1. We have one remark concerning results presentation: histograms and graphs are not nicely presented and could be improved. Y axis legend and value (12,000µm should be 12µm or 8,4 should be 8.4), or unnecessary horizontal lines, or absence of significant difference indications over histograms...

Thank you for the remark, the legend on the figures 1 and 4 has been corrected and unified.

Reviewer 3 Report

In this work, the Authors investigated bacterial Adhesion on commercially available Prosthetic and Orthotic Materials Surfaces. The work seems interesting, applicable, and well-written. Therefore, I would suggest accepting the manuscript Minor revisions. The reasons are mentioned below:

  1. Please check all, grammatical errors, and typos.
  2. All the bacteria names must be italic.
  3. The author in the introduction in some places mentioned “biofilm” for example in line 54. Please explain what is the concept of biofilm here?
  4. One time mentioning the complete name of bacteria is enough does not need to repeat them in this way “Staphylococcus (S.) aureus and Staphylococcus (S.) epidermidis” anymore. It can be easily written for example S.aureus
  5. Figures 6-8, the caption and images should refer similarly, not “A and a” or “B and b”
  6. The captions of figure 9 must rewrite.
  7. In line 308-311 it’s written: “Due to the uneven and highly diverse surfaces (porosity, surface features), the SEM micrographs were chosen over dye staining methods to ensure that the correct number of bacteria was counted on each micrograph and preferential attachment points evaluated at the same time. In Figure 5 surface coverage is shown for different materials.”

This is statement is not scientifically correct because for example Confocal microscopy can be used for all kinds of morphologies. Therefore, having uneven and highly diverse surfaces cannot be a reasonable reason for using SEM. I want the paragraph corrected, and the author explains why they claim for counting bacteria SEM is more precise than other available techniques.

Author Response

Reviewer 3:

  1. Please check all, grammatical errors, and typos.

Thank you for the reminder. Manuscript has been checked for grammatical errors and other typos.

  1. All the bacteria names must be italic.

This has been addressed.

  1. The author in the introduction in some places mentioned “biofilm” for example in line 54. Please explain what is the concept of biofilm here?

The “biofilm” in this sense is a continuous, single layer bacterial film that stretches across the material surface.

  1. One time mentioning the complete name of bacteria is enough does not need to repeat them in this way “Staphylococcus (S.) aureus and Staphylococcus (S.) epidermidis” anymore. It can be easily written for example S.aureus

It has been addressed. Thank you for the reminder.

  1. Figures 6-8, the caption and images should refer similarly, not “A and a” or “B and b”

Thank you for the remark, the formatting has been addressed.

  1. The captions of figure 9 must rewrite.

The caption has been expanded to: “SEM micrographs of (A) PMMA, (B) EVA/LDPE with Ag nanoparticles, (C) silicone and (D) TPE surfaces with adhered S. epidermidis.”

  1. In line 308-311 it’s written: “Due to the uneven and highly diverse surfaces (porosity, surface features), the SEM micrographs were chosen over dye staining methods to ensure that the correct number of bacteria was counted on each micrograph and preferential attachment points evaluated at the same time. In Figure 5 surface coverage is shown for different materials.”

This is statement is not scientifically correct because for example Confocal microscopy can be used for all kinds of morphologies. Therefore, having uneven and highly diverse surfaces cannot be a reasonable reason for using SEM. I want the paragraph corrected, and the author explains why they claim for counting bacteria SEM is more precise than other available techniques.

A keen observation. The choice for using SEM is actually two-fold: while uneven surface is indeed problematic, the main problem we encountered with our set of materials was the leakage of the staining dyes into the material itself. We did perform the optical measurements and found the results inconclusive with high scattering of the results due to the difficulties of separating the oversaturated background signal from the bacteria themselves. A sentence was added to the manuscript to further clarify our choices.